# Modern Approach to Melanoma Adjuvant Treatment with Anti-PD1 Immune Check Point Inhibitors or *BRAF/MEK* Targeted Therapy: Multicenter Real-World Report

**DOI:** 10.3390/cancers15174384

**Published:** 2023-09-01

**Authors:** Joanna Placzke, Magdalena Rosińska, Paweł Sobczuk, Marcin Ziętek, Natasza Kempa-Kamińska, Bożena Cybulska-Stopa, Grażyna Kamińska-Winciorek, Wiesław Bal, Jacek Mackiewicz, Łukasz Galus, Manuela Las-Jankowska, Michał Jankowski, Robert Dziura, Kamil Drucis, Aneta Borkowska, Tomasz Świtaj, Paweł Rogala, Katarzyna Kozak, Anna Klimczak, Paulina Jagodzińska-Mucha, Anna Szumera-Ciećkiewicz, Hanna Koseła-Paterczyk, Piotr Rutkowski

**Affiliations:** 1Department of Soft Tissue/Bone Sarcoma and Melanoma, Maria Skłodowska-Curie National Research Institute of Oncology, 02-781 Warsaw, Poland; 2Department of Computational Oncology, Maria Sklodowska-Curie National Research Institute of Oncology, 02-781 Warsaw, Poland; 3Division of Surgical Oncology, Department of Oncology, Wroclaw Medical University, 53-413 Wroclaw, Poland; 4Department of Clinical Oncology, Wroclaw Comprehensive Cancer Center, 53-413 Wroclaw, Poland; 5Department of Clinical Oncology, Maria Sklodowska-Curie National Research Institute of Oncology, 31-115 Kraków, Poland; 6Skin Cancer and Melanoma Team, Maria Sklodowska-Curie National Research Institute of Oncology, 02-781 Warsaw, Poland; 7Department of Medical and Experimental Oncology, University of Medical Sciences, 61-701 Poznan, Poland; 8Department of Clinical Oncology, Ludwik Rydygier Collegium Medicum, Nicolaus Copernicus University and Oncology Centre, 85-094 Bydgoszcz, Poland; 9Department of Oncological Surgery, Ludwik Rydygier Collegium Medicum, Nicolaus Copernicus University and Oncology Centre, 85-094 Bydgoszcz, Poland; 10Department of Clinical Oncology, Holy Cross Cancer Center, 25-734 Kielce, Poland; 11Department of Surgical Oncology, Medical University of Gdansk, 80-308 Gdańsk, Poland; 12Department of Pathology, Maria Sklodowska-Curie National Research Institute of Oncology, 02-781 Warsaw, Poland

**Keywords:** melanoma, adjuvant treatment, targeted therapy, immune check point inhibitors, early-stage melanoma, sentinel node biopsy, lymphadenectomy in melanoma, surgery de-escalation in melanoma

## Abstract

**Simple Summary:**

Performing this real-world analysis, we intended to see if postoperative (adjuvant) systemic treatment outcomes in a Polish melanoma patient population are comparable to the results from international trials based on which the treatment was registered worldwide. We intended to provide evidence on the efficacy and safety of postoperative melanoma treatment from everyday practice. We have shown that the type of surgical procedure on the lymph nodes prior to adjuvant treatment does not influence the outcome of that treatment. Our results support a de-escalation of surgery approach in melanoma patients. We support the value of adjuvant treatment for melanoma patients selected according to new guidelines implemented in parallel to the registration process. Analyzing the recorded side effects of adjuvant systemic treatment in our group of patients, we have noticed that severe complications worsen survival, giving us an indication not to treat by all means despite toxicity, particularly since it is a complementary to surgery treatment.

**Abstract:**

Background: The landscape of melanoma management changed as randomized trials have launched adjuvant treatment. Materials and Methods: An analysis of data on 248 consecutive melanoma stage III and IV patients given adjuvant therapy in eight centers (February 2019 to January 2021) was conducted. Results: The analyzed cohort comprised 147 melanoma patients given anti-PD1 (33% nivolumab, 26% pembrolizumab), and 101 (41%) were given dabrafenib plus trametinib (DT). The 2-year overall survival (OS), relapse-free survival (RFS), and distant-metastases-free survival (DMFS) rates were 86.7%, 61.4%, and 70.2%, respectively. The disease stage affected only the RFS rate; for stage IV, it was 52.2% (95% CI: 33.4–81.5%) vs. 62.5% (95% CI: 52.3–74.8%) for IIIA-D, *p* = 0.0033. The type of lymph node surgery before adjuvant therapy did not influence the outcomes. Completion of lymph node dissection cessation after positive SLNB did not affect the results in terms of RFS or OS. Treatment-related adverse events (TRAE) were associated with longer 24-month RFS, with a rate of 68.7% (55.5–84.9%) for TRAE vs. 56.6% (45.8–70%) without TRAE, *p* = 0.0031. For TRAE of grade ≥ 3, a significant decline in OS to 60.6% (26.9–100%; *p* = 0.004) was observed. Conclusions: Melanoma adjuvant therapy with anti-PD1 or DT outside clinical trials appears to be effective and comparable with the results of registration studies. Our data support a de-escalating surgery approach in melanoma treatment.

## 1. Introduction

Melanoma treatment has undergone a revolution in recent years. We have observed a shift in the early-stage treatment paradigm: de-escalation of surgery and increased use of adjuvant (and supposedly also neoadjuvant) systemic treatment. From 2016 to 2017, based on DECOG and MSLT-II randomized phase III trials, there exists strong evidence to omit completion lymphadenectomy (CLND) after positive sentinel lymph node biopsy (SLNB) [1,2]. SLNB, introduced by Morton [3], in addition to its therapeutic relevance, determines nowadays the need for adjuvant systemic therapy, which has become a standard of care for patients with high-risk resected melanoma of stage III and IV [4,5,6].

In 2017, the American Joint Committee on Cancer (AJCC) brought the updated Melanoma Staging System–Eighth Edition (AJCCv.8) to light [7]. In 2018, systemic adjuvant therapies with immune check point inhibitors (ICI): nivolumab and pembrolizumab and targeted BRAF/MEK inhibitors (BRAF/MEKi): dabrafenib plus trametinib were officially introduced in Europe based on the positive results of randomized phase III clinical trials: Checkmate-238 (for nivolumab vs. ipilimumab), Keynote-054 (for pembrolizumab vs. placebo), and Combi-AD (for dabrafenib plus trametinib vs. placebo) [8,9,10]. The registration of those therapies was based on clinical trials carried out according to inclusion criteria established before the new era; this involved patients with stage III-IV melanoma according to the AJCC v.7 (not 8) staging system with obligatory performance of CLND [9,10,11,12]. As melanoma adjuvant treatment established itself with increasing use and longer patient follow-up in everyday clinical practice, some questions have arisen, and uncertainties have emerged. Melanoma adjuvant treatment prolongs survival without relapse, thus avoiding its morbidity, but it has no proven impact on overall survival. It is potentially curative, but without an effect on survival in the entire population, it raises the topic of risk stratification for decision making. We need to search for predictive markers and weigh risks versus benefits in order to avoid overtreatment of those who can be cured only by surgery, while also having in mind possible chronic or permanent side effects and avoiding worsening quality of life for those who would benefit more from waiting for the recurrence strategy [6].

In our analysis we addressed some of those questions and uncertainties. First of all, we aimed to evaluate the effectiveness, safety and tolerability of adjuvant melanoma treatment in Polish patient population. Secondary, we intended to see if real-world results of this treatment in our patient population, not exactly represented in adjuvant registration randomized controlled trials, are comparable to the results from those studies. In addition, we took under consideration possible prognostic and predictive markers. We proposed a prognostic model and evaluated some factors as type of surgery prior to adjuvant treatment or drug used for adjuvant treatment (e.g., anti-PD1 vs. BRAF/MEKi for selected *BRAF/*+**/** melanoma patients) which could influence adjuvant treatment results. Finally, we have investigated whether adjuvant systemic treatment outcomes correlate with occurrence of related to treatment side effects.

## 2. Materials and Methods

### 2.1. Patient Cohort and Inclusion Criteria

The subject of our analysis was a group of consecutive patients identified with stage III–IV resected melanoma who were given the adjuvant treatment between April 2019 and January 2021 in 8 Polish melanoma reference centers. The inclusion criteria were: histologically confirmed melanoma (cutaneous acral, mucosal, or melanoma of unknown primary site were accepted), >18 years of age, resected R0 stage III or IV melanoma by AJCCv.8 (CLND after positive SLNB was not mandatory), *BRAF* status determined (with no need for *BRAF* variant assessment), at least one given cycle of adjuvant systemic treatment with immune check point inhibitors (ICI), anti-PD1 antibodies (nivolumab or pembrolizumab or targeted therapy), and BRAF/MEK inhibitors (dabrafenib plus trametinib (DT)). Schedules and dosage were according to the products’ characteristics. The data cut-off point was 31 January 2022. Strict follow-up was performed according to the description of the National Drug Program.

### 2.2. Collected Covariates

Collected data included: patient demographics; primary melanoma characteristics including location, histologic subtype, *BRAF* mutation status, Breslow tumor thickness, and presence of ulceration and microsatellites; nodal characteristics including date, type, and result of nodal surgery performed (sentinel lymph node biopsy, SLNB; completion lymph node dissection, CLND; therapeutic lymph node dissection, TLND), number of nodes assessed, number of nodes positive for metastatic disease, diameter of nodal metastasis; American Joint Committee on Cancer (AJCC) version 8 staging; description of adjuvant therapy (type of drug used for systemic treatment); duration of adjuvant therapy; reason for adjuvant therapy cessation; appearance of adverse events (TRAE); grade of TRAE; treatment for TRAE; recurrence date; location of recurrence; type of additional therapy for recurrence; and date of death or last follow-up, whichever was eligible.

### 2.3. Analysis Plan

We determined the demographics, disease and treatment characteristics, and rate of toxicity according to adjuvant management (immunotherapy vs. targeted therapy). In different subgroups, we measured the time to event outcomes at 2 years: recurrence-free survival (RFS)—the time from the date of adjuvant treatment start to any recurrence, distant-metastasis-free survival (DMFS)—the time from adjuvant treatment commencement to distant metastasis appearance and overall survival (OS)—the time from adjuvant treatment commencement to death or last observation, regardless of disease recurrence.

### 2.4. Statistical Methods

The data were pseudonymized. Categorical variables were described as a proportion with the percentage, and continuous variables were described as a mean and standard deviation or a median with an interquartile range. The 2-year relapse-free survival (RFS), distant-metastases-free survival (DMFS), and overall survival (OS) were estimated using the Kaplan–Meier method. A log-rank test was used to examine differences between subgroups. Multivariate analysis was performed using the Cox Proportional-Hazards Model. Patients’ clinical data and outcomes were analyzed through the use of “R” Software v.2021, in particular with the application of “survival”, “survminer,” and “rms” packages [13].

### 2.5. Ethical Statement

This retrospective, noninterventional study was conducted according to the guidelines of the Declaration of Helsinki and approved by the Institutional Ethics Committee of the Maria Skłodowska-Curie National Research Institute of Oncology; ethical board approval number 27/2018 dated 19 April 2018. Prior to any given treatment, the patients signed an informed consent form for the treatment and a consent form allowing the usage of their data for scientific purposes.

## 3. Results

### 3.1. Patient and Tumor Characteristics

Overall, we identified 248 patients who met the inclusion criteria. The median follow-up was 18.9 months (95% confidence interval CI: 17.9–20.0). There were 47% females, with a median age of 57 years (IQR: 45–68; range: 20–88); in total, 54% of patients had comorbidities, and 65% were of WHO 0 performance status. The median Breslow thickness was 3.2 mm (IQR: 1.8–6.0 mm), ulceration was present in 48% of patients, and 63% (155) of patients had a positive *BRAF* mutational status. The stage distribution according to AJCCv.8 was as follow: IIIA 6%, IIIB 23%, IIIC 56%, IIID 4%, and IV in 9% of patients. In total, 87% of the melanomas were cutaneous with exclusion of acral subtype, 3% were cutaneous/acral, and only three cases were mucosal. Patient and tumor characteristics are summarized in Table 1.

### 3.2. Local Therapy

#### 3.2.1. Surgical Treatment before Adjuvant Systemic Therapy

For the purpose of analysis, all patients were clustered into three groups according to the type of surgery before adjuvant treatment: surgery without recurrence (primary melanoma and nodes: positive sentinel lymph node biopsy (SLNB) +/− completion lymph node dissection (CLND)), surgery after local recurrence (therapeutic lymph node dissection (TLND) +/− in-transit resection), and surgery after distant recurrence (resected M1) (Table 2). Within the primary surgery group (resection of primary melanoma and/or nodes), there were patients who underwent TLND without SLNB (33, 25.4%), those who underwent CLND after positive SLNB (63, 48.5%), and those who had positive SLNB but no CLND afterwards (34, 26.2%). 

There were 174 (70%) patients with intervals longer than 13 weeks from the last operation to adjuvant treatment start.

#### 3.2.2. Radiotherapy

Adjuvant radiotherapy was recorded in 15 (6%) analyzed patients (10 patients from ICI and 5 patients from DT subgroups).

### 3.3. Systemic Treatment

Applied adjuvant therapy consisted of nivolumab in 82 patients, pembrolizumab in 65 patients, and a combination of dabrafenib and trametinib (DT) in 101 patients (Table 2). In our cohort, adjuvant immunotherapy was applied in 54 (37%) melanoma *BRAF*-positive and 93 (63%) *BRAF*-negative patients. At the time of cut-off, 184 (74%) patients finished adjuvant treatment, with 107 (71%) on anti-PD1 and 77 (76%) on DT. One-year adjuvant therapy was completed as scheduled in 60% of patients (54 (50%) on immunotherapy and 57 (75%) on targeted therapy). Therapy was discontinued due to treatment-related adverse events (TRAE) in 24 patients (13%) and due to relapse in 39 patients (21%).

The overall incidence of TRAE was 46%, which was higher in DT in comparison to the anti-PD1 group at 58% and 39%, respectively (Table 3). The most common was grade two TRAE at 21%. Grade three TRAE was observed in 23 (9.3%) patients. In total, 29% of the overall TRAE led to dosage reduction or treatment discontinuation.

### 3.4. Survival Analysis

#### 3.4.1. Registered Events

At a median follow-up of 13.9 months (average: 15.0 months; SD = 6.5), 61 (25%) patients experienced disease recurrence. There were 46 (31%) relapses in the ICI group and 15 (15%) in the DT group (Table 4). Most of the recurrences (64%) occurred during adjuvant treatment, particularly in the ICI group. Overall, 75% of all recurrences occurred in the immunotherapy arm. The most common locations of recurrence were metastases in visceral organs (34%), the skin, subcutaneous tissue, and distant nodes (25%), as well as local in-transit or scar recurrences (18%). The brain was a site of relapse in 8% of patients. Most of the relapses occurred in patients with stage IIIC melanoma. During the minimum observation period of 12.2 months (range up to 36.4 months), 11 (18%) relapses led to death.

There were 13 (5%) deaths in total. Of these deaths, 69% of them were due to melanoma relapse, 31% were from unknown causes, and two deaths (15%) were likely due to TRAE. The distribution of deaths between the treatment groups (ICI vs. DT) was equal, with 5% in each group. The restricted mean survival during the first 24 months was 23.1 months.

We have observed relapses and fatal events to be distributed evenly between the groups split by criterion of less or equal to 13 weeks from the last operation to adjuvant start, with 19 (26%) relapses and 7 (9%) deaths in the ≤13-week group and 43 (25%) relapses and 11 (6%) deaths in the >13-week group.

#### 3.4.2. Overall and Recurrence-Free Survival

The probabilities of 2-year overall survival (OS), relapse-free survival (RFS), and distant-metastases-free survival (DMFS) were 86.7%, 61.4%, and 70.2%, respectively, in the entire study population of adjuvant-treated melanoma patients; these data are summarized in Table 4 and shown graphically in Figure 1. Medians were not reached at the time of analysis.

#### 3.4.3. Survival Depending on Patient and Melanoma Characteristics

There was no significant association related to OS and RFS in the adjuvant treatment results with age, gender, comorbidities, or any other patient characteristics, as shown in Appendix B and Table A5 and Table A6. OS and RFS results of adjuvant treatment within our multicenter cohort were consistent; no significant differences in survival results were observed among eight melanoma centers (Appendix B, Figure A1).

The disease stage before adjuvant treatment did not affect overall survival, but the RFS rate was significantly smaller for stage IV vs. stage IIIA-D at 52.2% (95% CI: 33.4–81.5%) vs. 62.5% (95% CI: 52.3–74.8%), *p* = 0.0033, respectively (Figure A2 in Appendix B).

Patients with *BRAF*/+/melanomas had a higher probability of 2-year RFS compared to melanoma *BRAF*/−/patients at 62.9% (95% CI: 49.4–80%) vs. 57.7% (95% CI: 46.4–71.7%), respectively, *p* = 0.023. The relationship was no longer visible in OS analysis (88.6% (95% CI: 80.1–97.9%) vs. 82.7% (95% CI: 67.5–100%), *p* = 0.74) (Figure A3 in Appendix B).

#### 3.4.4. Survival Association with Surgical Procedure before Adjuvant and the Adjuvant Type

The type of surgery before adjuvant treatment of melanoma did not affect OS. Surgery of distant metastases had a negative effect only on the probability of relapse-free survival (*p* = 0.0062), although the difference decreased at the end of the 2-year observation period, with a 2-year RFS of 52.2% (95% CI: 33.4–81.5%) in the M1 group vs. 58.8% (95% CI: 45.3–76.5%) in the local recurrence group (Figure 2, Table 4). Of note, there was no significant difference in overall survival across the studied groups.

The type of drug used for adjuvant systemic therapy did not significantly influence the probability of 2-year OS in the whole group (*p* = 0.6318), as shown in Table 4 as well as in Table A5 and Table A6 Appendix B. A difference in RFS rates in relation to the used drugs was significantly unfavorable for anti-PD1 adjuvant treatment, especially at the beginning of the observation period, which is graphically presented in Figure 3. The 2-year RFS was 56.1% (45.6–69.1%) and 65.9% (48.4–89.7%) for anti-PD1 and BRAF/MEKi, respectively. This difference was evident both in the whole group and in the *BRAF*/+/group (Figure 4). The *BRAF/*+/melanoma patients had higher rates of 2-year recurrence-free survival (RFS) in the entire cohort, as well as in the DT-treated group, compared to those treated with anti-PD1 (Figure 4 and Figure A3 in Appendix B). The effects of TRAE on OS and RFS were similar across the subpopulations defined by the surgery before adjuvant treatment (Table A9, Appendix B).

#### 3.4.5. Survival According to Treatment-Related Adverse Events (TRAE)

We have observed that the occurrence of TRAE was associated with significantly longer RFS in entire cohort. The 24-month RFS rate in patients with TRAE was 68.7% (55.5–84.9%) vs. 56.6% (45.8–70%) in patients without TRAE; *p* = 0.0031 (Figure 5). When analysis was performed in immunotherapy and targeted therapy groups separately, it appeared that the effect was basically restricted to RFS in a group treated with anti-PD1 (Figure 6 and Table A5, Appendix B). Survival probability without relapse was higher when adverse events occurred during or after anti-PD1 treatment (Figure 6). Treatment termination due to TRAE was also associated with longer RFS and did not affect OS (Figure A4 and Figure A5, Appendix B).

According to our findings, grade ≥ 3 TRAE significantly worsened the overall survival prognosis (although not relapse-free survival). The 2-year OS rate decreased to 60.6% (26.9–100%), *p* = 0.004, as presented in Figure A6 and Table A6 in Appendix B.

Analysis of TRAE was not the focus of the current analysis, and we did not collect sufficient data to be able to investigate the topic of the differences in treatment dose intensity between the TRAE group and the non-TRAE group. We summarized the available data in Appendix C.

### 3.5. Prognostic Model

We have developed a prognostic model indicating that the difference in RFS between the two treatments is limited to the first months after the start of treatment, when there is a higher risk of progression for patients treated with anti-PD1 (HR 5.8, 95% CI 1.3–24.9; *p* = 0.0183), as shown in Table A3 in Appendix A. At longer follow-up (i.e., over 18 months), there is a reversed tendency (higher hazard for BRAF/MEK inhibitors), and the relationship disappears graphically, but it was not statistically significant, possibly due to a small number of cases with longer follow-up. To explore the possible impact of inter-site variability, a shared frailty model was fitted with the consistent results. The difference is not seen in the OS analysis in the above-mentioned groups.

### 3.6. Subgroup Analysis of Primary/LND Group

An additional subgroup analysis concerned the predictors of overall and relapse-free survival in the primary melanoma group (*n* = 130). We have observed a negative association of melanoma thickness above 4 mm with adjuvant treatment outcomes. Primary T4a-b melanomas had a significantly worse 2-year OS rate of 66% (41.3–100%) in comparison to 97.8% (93.6–100%) in ≤pT2 and 92.3% (78.9–100%) in pT3; *p* = 0.02. The respective 2-year RFS rates were 54% (37.8–77.1%), 88.4% (78–100%), and 59.3% (30.9–100%) (Table A7 and Table A8, Figure A7 in Appendix B). The absence of ulceration positively impacted the 2-year OS rate (100% vs. 70.6% (52.2–95.6%); *p* = 0.0114)), which is presented in Figure A8 in Appendix B. Acral melanomas had a significantly worse outcome of adjuvant treatment with respect to OS and RFS rates, as shown in Figure A9 in Appendix B.

The type of lymph node surgery before adjuvant treatment (SLNB ± CLND or TLND) also did not influence the outcome with respect to OS and RFS in our cohort, as shown in Figure 7 and in Table A7 and Table A8 in Appendix B. In particular, not performing CLND after positive SLNB did not affect the results of adjuvant treatment in terms of RFS or OS. In the CLND group, compared to the no CLND group, the 2-year RFS was worse (56.5% (37.7–84.9%) vs. 80% (51.6–100%)), although the difference was not statistically significant.

## 4. Discussion

The locally advanced melanoma treatment paradigm is evolving. Adjuvant systemic treatment for patients with early-stage III-IV melanoma after radical resection became available worldwide since 2018 based on the results of randomized phase III clinical trials: Checkmate-238 (nivolumab vs. ipilimumab), Keynote-054 (pembrolizumab vs. placebo), and Combi-AD (dabrafenib plus trametinib vs. placebo) [8,9,10]. Additionally, melanoma adjuvant therapy’s scale of relapse risk reduction appeared to be the one of the highest among adjuvant treatments of solid tumors [6].

### 4.1. Approach to Adjuvant Melanoma Treatment Inclusion Criteria

In the meantime, for designing, conducting, and registering adjuvant clinical trials, a new approach to completion lymphadenectomy after positive sentinel lymph node biopsy and an actualized version of the AJCC Melanoma Staging System Eighth Edition were established. Polish patients started to receive adjuvant treatment since 2019 based on individual applications for emergency access to medical technologies. Inclusion criteria for melanoma adjuvant therapy in the Polish patient population were adjusted to those from registration trials but in adherence to the updated guidelines and to the new situation at some point. The main inclusion criteria adjustments for adjuvant therapy against all three registration trials included: staging―the AJCCv.8 classification was used (in contrast to AJCCv.7 in registration trials), non-mandatory CLND, a wider defined interval to adjuvant treatment start (restricted to 12–13 weeks from randomization in trials), and permission for radiotherapy. Patients with acral cutaneous subtype and mucosal melanoma could be enrolled (as in Keynote-054, but not in Checkmate-238 nor in Combi-AD). Resected in-transit metastases were allowed (as in Checkmate-238 and Combi-AD, but not in Keynote-054). Stage IIIA (as in Keynote-054 and Combi-AD, but not in Checkmate-238) and resected stage IV (as in Checkmate-238 but not in Keynote-054 nor in Combi-AD) were eligible. Adjuvant treatment with targeted therapy was not restricted to patients with melanoma with *BRAF V600E* or *V600K* mutations (as in the Combi-AD trial).

In light of a new approach toward melanoma staging and a lack of indications for completion of lymphadenectomy, we intended to provide evidence on the efficacy and safety of adjuvant melanoma treatment for patients not quite in line with those strictly meeting the inclusion criteria of randomized clinical trials. Since 2016, based on the Multicenter Selective Lymphadenectomy Trial II (MSLT-II), and since 2017, based on the German Dermatologic Cooperative Oncology Group trial (DeCOG-SLT), SLNB followed by CLND is no longer a standard of care for all patients with SLN-positive disease [1,2]. In those trials, no improvement in melanoma-specific survival (MSS) from CLND was demonstrated. Surgeons do not perform routine CLND after positive SNLB anymore, and they refer patients to adjuvant systemic treatment [14,15]. Omitting CLND leaves us without precise information on local disease burden. We are left with an overrepresented IIIA stage that has different risks than the former IIIA from the CLND era. Stages IIIB and IIIC might be underdiagnosed while CLND is no longer performed, and some of metastases stay occult. While AJCCv.8 staging was applied in the era of postponed CLND, prognosis estimates for IIIA and IIIB subgroups worsened due to occult disease burden in patients not having CLND after positive SLNB. Adjuvant therapy benefits across the stage III subgroups by AJCCv.8 classification may be even more profound as prognosis for the real-world patients is worse in comparison to the AJCC report, as shown in a German study based on the European population [16]. Adjuvant treatment outcomes in our dataset are in line with those of the Keynote-054 trial, where RFS benefits were confirmed for stage III subgroups adjusted according to AJCCv.8 [17].

The impact of the CLND approach shift on melanoma adjuvant treatment results was never investigated in a prospective manner, and it was only investigated in real-world datasets of national registries or smaller multicenter analysis, such as ours. In our study, we examined a group of patients, all of whom have received adjuvant treatment, regardless of nodal resection status (SLND+ and CLND− or CLND+ were allowed). We have shown that patients without CLND have the same adjuvant treatment outcomes as patients after CLND for positive SLNB. As completion lymphadenectomy influenced only the local control, not melanoma-specific survival, a possible lack of local control is not the reason for events during adjuvant therapy (the rate of distant metastases at relapse was 82% in our cohort) [1,2]. Our data on adjuvant treatment of melanoma patients without CLND support the rationale for de-escalation of surgery. CLND can be all the more omitted as adjuvant treatment is applied. Melanoma stage III patients with positive sentinel nodes benefit from adjuvant systemic treatment and not from surgery escalation [18]. In accordance with other findings, we advocate for the principle of delayed TLND in the case of local lymphatic recurrence [19,20,21]. In our cohort, TLND after the surveillance period did not compromise melanoma adjuvant treatment results. Eventually, any type of lymph node surgery before adjuvant treatment (SLNB ± CLND or TLND) did not influence the outcome with respect to OS and RFS in our analysis; only the resection of distant metastases had a negative impact due to the worse prognosis and the anticipated, less pronounced effect of adjuvant therapy in this group of patients [22].

A modern approach to CLND has changed our attitude toward SLNB, as nowadays, it serves as a therapeutic and staging procedure, qualifying patients for adjuvant treatment [4]. SLNB also offers a cure in itself, providing the possibility of local lymphatic basin control, and some patients could be cured only with the procedure of SLNB surgery [23,24]. According to some researchers, SLNB can eradicate the regional disease in three quarters of patients [4]. In our study, not performing SLNB with late TLND did not have a negative impact on RFS or on OS while applying adjuvant treatment.

### 4.2. Prognostic and Predictive Tools

The largest prospective randomized trial on long-term regional basin control after removal of a positive SNLB has pointed to the positive prognostic association with basin control of such variables as younger age, thinner primary melanoma, axillary basin, and metastasis diameter less than 1 mm with area less than 5% [23]. There is a need to determine who might be cured by SLNB alone (low-risk group) and who is at high risk (at risk of nonsentinel lymph node metastases) and might benefit from adjuvant therapy [18]. There are being developed various prognostic tools and nomograms trying to determine who is at high risk of nonsentinel lymph node metastases (NSLN) to determine who needs adjuvant therapy. Most of them elaborate extensively on clinicopathological features. A recent study performed by a Dutch–American group has shown that currently available predictive models are not efficient and not validated for single positive SLNB, and they show limited performance in this subgroup of patients [25]. Their study analyzed the specific patient group with a single positive SLN, and they did not find any model that would identify patients at high risk of NSLN in a group of single positive SLN. In their research, it appeared that predictions heavily relied on the number of positive sentinel nodes after SLNB. They found out that the rule of thumb (patients at high risk have multiple positive SLN) performed as well as their newly developed model based on age, Breslow thickness, mitotic rate, largest SLN metastasis diameter, and the number of positive SLN (single vs. more than one node). Accuracy in making predictions of the rule of thumb was comparable to the best-performing Bertolli nomogram based on Breslow thickness, SLN disease burden/diameter, and number of positive SLN [26].

Patients with stage III melanoma have a high risk of distant metastases, and prognostic tools for identifying those at high risk of them are needed. The problem of survival lies within a spectrum of distant metastases, and translational research implementing information from clinical, histopathological, serological, and molecular biomarkers would be of predictive and prognostic value [27,28]. Researchers from the Dutch–American group assume that exploitation of the clinicopathological features of SLN has reached its peak [25]. Scientists are looking closely at early-stage melanoma given that the biggest rate of deaths is attributed to thin melanomas (23%), while thick or metastatic-at-presentation melanomas are responsible for 14% and 16% of deaths, respectively [29]. It is of great importance to search for predictive tools to define patients of high risk of recurrence or death who would benefit the most from additional treatment interventions. Melanoma genetics, tumor microenvironment, and serological markers are being explored with a view to identify those at high risk of distant metastases for whom more than surgery, e.g., adjuvant treatment, should be offered [30,31,32,33]. There are studies on resistance to treatment dealing with issues of minimal residual disease and its heterogeneity, drug-tolerant phenotypes, genomics, and molecular aberrations [34,35,36,37]. In our study, we have also looked at the spectrum of lower susceptibility to adjuvant therapy, and we have found that such variables associated with primary melanoma, such as thickness, ulceration, and acral cutaneous and mucosal melanoma subtypes, were negatively associated with overall survival in our cohort in univariate and multivariate analyses.

### 4.3. Systemic Treatment Impact

Prediction tools guiding our decisions on adjuvant treatment start and on drug selection are important in consideration of a possible significant share of chronic side effects generated by adjuvant therapy, some of them permanent [38]. In our study, there was less than 10% of treatment-related adverse events (TRAE), and 13% of patients discontinued the treatment due to TRAE; these results are comparable to those from randomized trials. Nearly thirty percent of overall TRAE led to dosage reduction or treatment discontinuation, and this gave us indirect information about the quality of life of melanoma patients during adjuvant treatment. Based on our results, we know that toxicities might be detrimental. Grade three toxicities significantly worsen OS in our real-world population, giving us a guideline not to treat by all means despite toxicity. On the other hand, the occurrence of TRAE or treatment termination due to TRAE were associated with longer RFS but not OS for the anti-PD1 treated group in our study. A similar association was observed by other researchers interpreting randomized data on nivolumab and pembrolizumab in adjuvant treatment [39]. We have not observed an association between the occurrence of TRAE and survival in the BRAF/MEKi treated group. In our cohort, patients with melanoma of *BRAF*-positive mutational status were treated both with BRAF/MEKi and anti-PD1 drugs. We have shown short-term better control of melanoma relapses when RAF/MEKi were used in patients with *BRAF*-mutated melanomas; however, beyond 2 years, long-term outcomes overlap with immunotherapy with anti-PD1 antibodies. This result probably depends mainly upon the different modes of action of available adjuvant drugs and sparks a discussion on the topic of appropriate adjuvant therapy selection for melanoma *BRAF*-positive patients. BRAF/MEKi provide a rapid response, while anti-PD1 antibodies act slowly and may achieve a durable response [40]. No direct comparison between adjuvant targeted therapy and immunotherapy for melanoma *BRAF*-positive patients has been performed. Therefore, eligibility for the adjuvant treatment and adjuvant drug selection should be carefully assessed while discussing pros and cons with the patient. Such criteria as risk versus benefit estimation, possibility of chronic and permanent toxicities, discontinuation rate, contraindications, and registration indications and disease burden should be taken into consideration [6].

### 4.4. Real-World Analysis in Comparison to Registration Trials

The results of our study show that a real-world cohort of melanoma patients not fully represented in phase III clinical trials benefit from adjuvant treatment. Our 2-year RFS of 61.4% is comparable to outcomes for adjuvant treatment arms in randomized trials; the estimated 2-year RFS rate in Checkmate-238 was around 62% for nivolumab, in Keynote-054 it was 68% for pembrolizumab, and in Combi-AD it was 67% for dabrafenib plus trametinib. OS at 2 years for 86.7% of our cohort is similar to the results from Combi-AD and Checkmate-238, which were estimated 91% and 87%, respectively. In all three registration trials, only RFS rates demonstrated significant benefit for experimental arms, with no OS benefit shown. The recurrence-free survival rate is a measure of all relapses, both distant and local. In registration trials, 30–60% of recurrences were distant [6]. In our study, 82% of relapses were distant, including one fourth of M1a recurrences. Improving the RFS rate limits disease spread and reduces the number of patients with advanced disease. It is important to reduce the number of patients with advanced-stage melanoma [6]. In adjuvant trials, RFS with a hazard ratio (HR) of 0.77 or less can be a surrogate for OS [41]. In randomized adjuvant trials, to which our results are comparable, the HR was around 0.6. We assume, therefore, that our adjuvant treatment outcome (of comparable RFS to that obtained in registration trials) is likely to translate into a gain in overall survival. OS benefit is hard to achieve, if only because of the subsequent lines of effective melanoma treatment nowadays.

Real-world analysis can help us better understand the details and dilemmas of novel treatment methods or approaches outside clinical trials, bring valuable information on safety and efficacy of a certain treatment, and could be used to inform decisions for new drug approvals, indications, or population expansions [42]. Our data and analysis on adjuvant treatment outcomes in melanoma patients, along with other research reports, show that adjuvant treatment of melanoma in the real-world population reduces the risk of recurrence, even outside the criteria from the registration clinical trials [43,44,45]. We have shown better short-term control of melanoma relapses when DT was used in patients with *BRAF*-mutated melanoma. However, after approximately two years, the long-term DT effect overlaps with immunotherapy outcomes for this group of patients.

### 4.5. Strengths and Limitations

The strength of our study is the analysis of real-world data outside clinical trials supporting the value of adjuvant treatment of melanoma patients with inclusion criteria tailored according to new guidelines implemented during the registration process. This is a unique and detailed study examining a group of patients, all of whom have received adjuvant treatment regardless of nodal resection status; it shows that the type of lymph node surgery before adjuvant treatment does not influence the outcome of that treatment. In light of a new classification and new approach to CLND, our study confirmed that postponing CLND was justified and fits into a new landscape of adjuvant treatment.

Limitations of our analysis are its retrospective design, the relatively small number of patients in each subgroup, and short follow-up. Longer observation is needed to gain detailed information about recurrences, their treatment, and the influence of TRAE on survival.

## 5. Conclusions

Multicenter data collected from across Poland confirm that a modern approach to adjuvant treatment of melanoma patients is well tolerated and offers promising early overall survival and recurrence-free survival comparable with the results of registration clinical trials. Adjuvant treatment of melanoma in the real-world environment reduces the risk of recurrence and hence reduces morbidity of stage III and IV melanoma patients in the landscape of updated AJCCv.8 staging classification and CLND evasion, while not fully in line with inclusion criteria from phase III adjuvant melanoma treatment registration trials. Adjuvant therapy for melanoma patients offers the opportunity to safely de-escalate surgical treatment without the risk of worsening prognosis. Further follow-up is needed to evaluate long-term toxicities and outcomes of melanoma adjuvant treatment in this population.

## Figures and Tables

**Figure 1 cancers-15-04384-f001:**
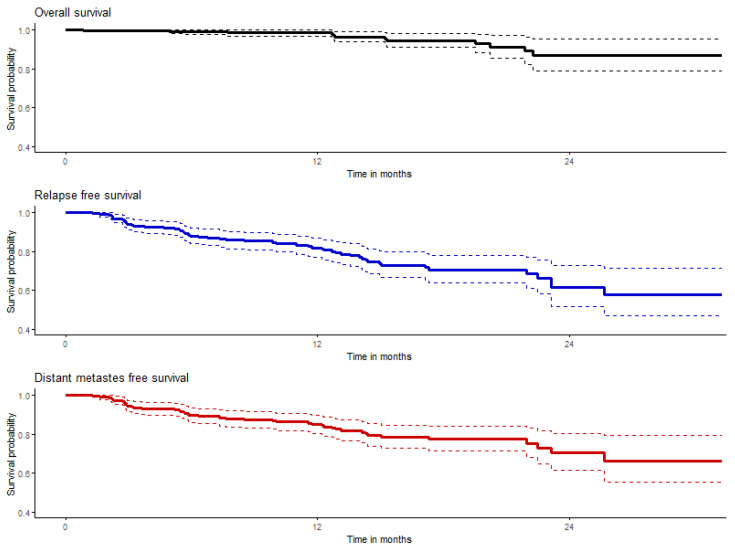
Graphs of overall (black), relapse (blue), and distant-metastases-free survival (red); colored solid lines represent estimated probabilities of survival and dashed lines represent their confidence intervals.

**Figure 2 cancers-15-04384-f002:**
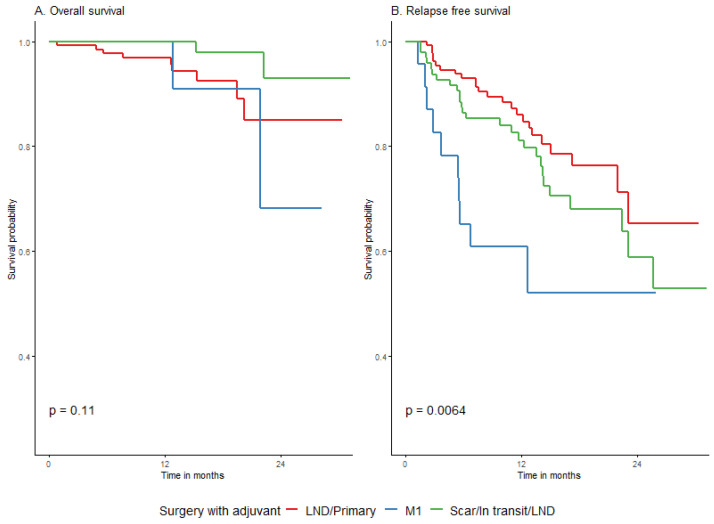
Impact of surgery before adjuvant on overall (**A**) and relapse-free (**B**) survival.

**Figure 3 cancers-15-04384-f003:**
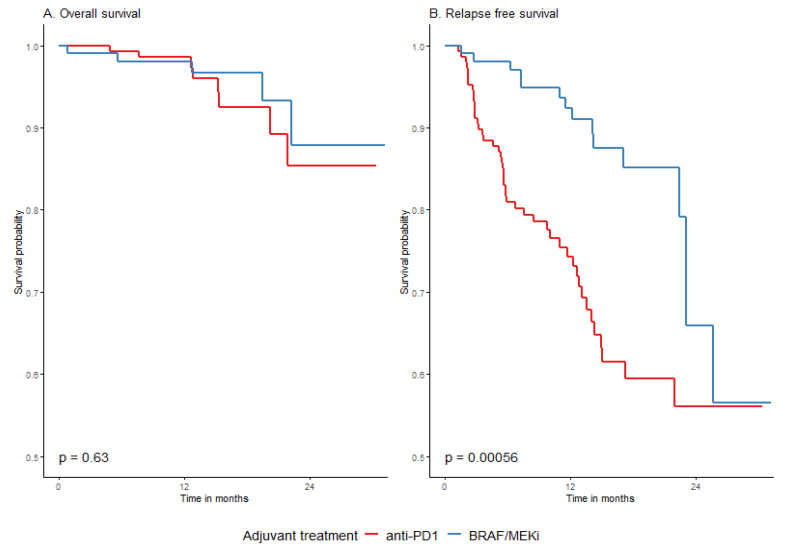
Overall (**A**) and relapse-free (**B**) survival in relation to the type of drugs used; DT vs. anti-PD1.

**Figure 4 cancers-15-04384-f004:**
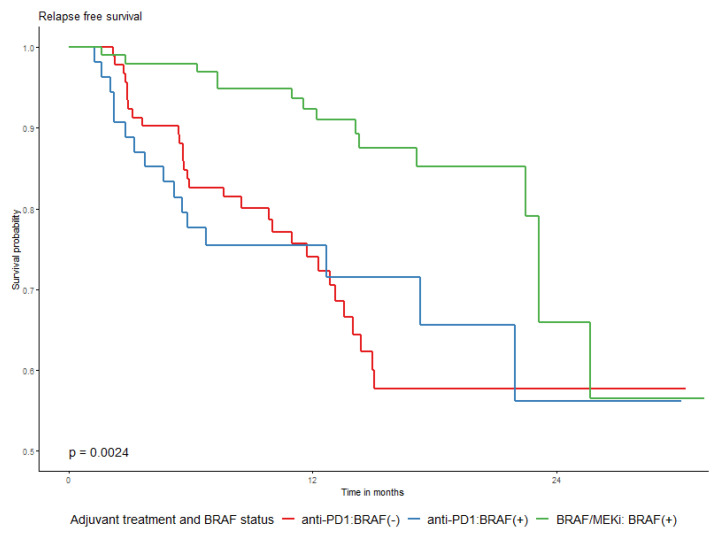
Relapse-free survival by *BRAF* mutational status and adjuvant drug used; anti-PD1 or BRAF/MEKi.

**Figure 5 cancers-15-04384-f005:**
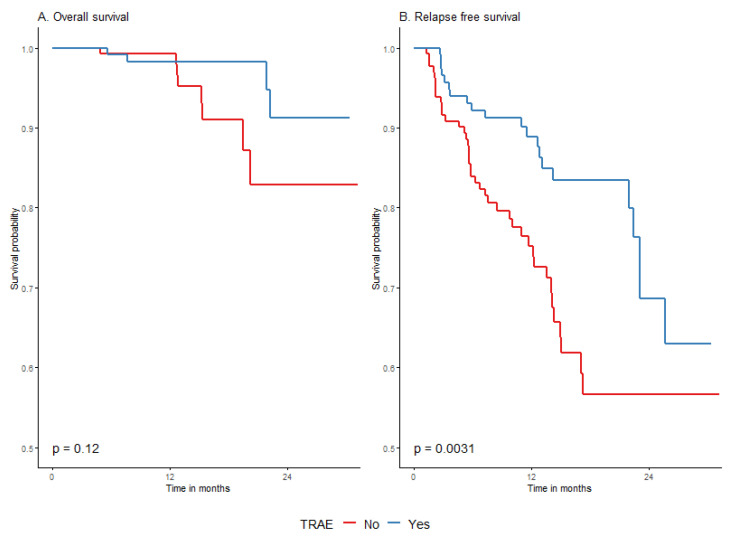
Overall survival (**A**) and relapse-free survival (**B**) by occurrence of TRAE.

**Figure 6 cancers-15-04384-f006:**
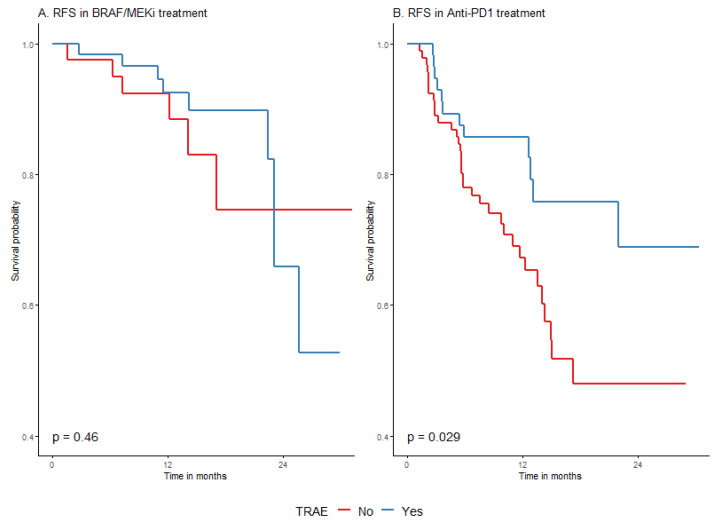
RFS by treatment-related adverse events (TRAE) in BRAF/MEKi (**A**) and anti-PD1 (**B**) groups.

**Figure 7 cancers-15-04384-f007:**
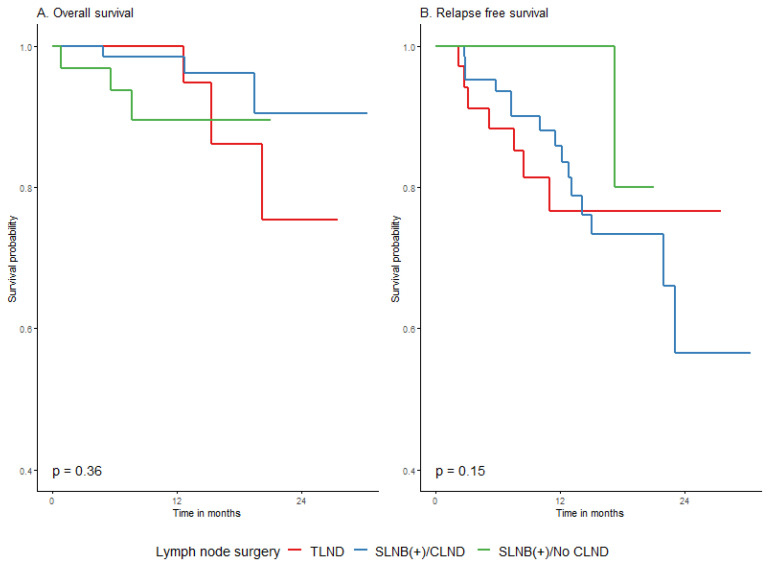
Impact of lymph node surgery prior to adjuvant treatment on overall (**A**) and relapse-free (**B**) survival in primary melanoma group.

**Table 1 cancers-15-04384-t001:** Demographic and primary tumor characteristics (*n* = 248).

Covariate	Group	Total	
		*n*	%
Sex	female	116	46.8
	male	132	53.2
Age group	<45	62	25
	45–54	41	16.5
	55–64	64	25.8
	65+	81	32.7
Melanoma type	skin	215	86.7
	acral	8	3.2
	mucosal	3	1.2
	unknown	22	8.9
*BRAF* mutational status	negative	92	37.1
	positive	155	62.5
	undocumented	1	0.4
AJCC8 primary tumor characteristics, pT	T0	21	8.5
	T1a–b	20	8.0
	T2a–b	43	17.3
	T3a–b	62	25
	T4a–b	93	37.5
	NA	9	3.6

**Table 2 cancers-15-04384-t002:** Adjuvant treatment.

Covariate	Group	Total	
		*n*	%
WHO performance status at treatment initiation	0	161	64.9
	1	86	34.7
	2	1	0.4
AJCC8 stage at initiation of adjuvant treatment	IIIA	16	6.5
	IIIB	56	22.6
	IIIC	139	56
	IIID	11	4.4
	IIIx	3	1.2
	IV	23	9.3
Group of indications for adjuvant	Primary surgery (primary melanoma and nodes)	130	52.4
	Resected distant metastases	23	9.3
	Resected local recurrence (scar/in transit/nodal)	95	38.3
Node surgery in the primary surgery group	SLNB (+), no CLND	34	26.2
	SLNB (+), CLND	63	48.5
	TLND	33	25.4
Adjuvant type	dabrafenib plus trametinib	101	40.7
	pembrolizumab	65	26.2
	nivolumab	82	33.1
Adjuvant treatment ended	no	64	25.8
	yes	184	74.2
Reason for ending of adjuvant treatment	(Treatment ongoing)	64	25.8
	As planned	111	44.8
	Other	4	1.6
	Progression	39	15.7
	Toxicity	23	9.3
	Undocumented	7	2.8

SLNB: sentinel lymph node biopsy; CLND: completion lymph node dissection; TLND: therapeutic lymph node dissection.

**Table 3 cancers-15-04384-t003:** Treatment-related adverse events.

Covariate	Group	Anti-PD1	DT	Total	*p*-Value
		*n* (%)	*n* (%)	*n* (%)	
Toxicities after treatment	Yes	56 (38.1)	59 (58.4)	115 (46.4)	0.001
	No	91 (61.9)	40 (39.6)	131 (52.8)	
	Undocumented	0	2 (2.0)	2 (0.8)	
Treatment ended due to toxicity	Yes	13 (8.8)	11 (10.9)	24 (9.7)	0.302
	No	91 (61.9)	68 (67.3)	159 (64.1)	
	(Treatment ongoing)	43 (29.3)	21 (20.8)	64 (25.8)	
	Undocumented	0	1 (1.0)	1 (0.4)	
Highest grade toxicity if toxicity was reported (*n* = 115)	1	23 (41.1)	14 (23.7)	37 (32.2)	0.041
	2	25 (44.6)	26 (44.1)	51 (44.3)	
	3	8 (14.3)	15 (25.4)	23 (20.0)	
	Undocumented	0	4 (6.8)	4 (3.5)	

Anti-PD1: nivolumab or pembrolizumab; DT: dabrafenib plus trametinib.

**Table 4 cancers-15-04384-t004:** Overall and recurrence-free survival.

		*n*	*n* Events ****	Restricted Mean * (SD)	24-Month Survival (95% CI)	*p*-Value
Recurrence-free survival (RFS)						
	Overall	248	61	19.5 (0.5)	61.4% (51.8–72.8%)	
Indication group	LND/Primary	130	24	20.5 (0.6)	65.4% (50.9–83.9%)	0.0064
	M1	23	10	15.2 (2.1)	52.2% (33.4–81.5%)	
	Scar/In transit/LND	95	27	19.2 (0.8)	58.8% (45.3–76.5%)	
Adjuvant group	Anti-PD1	147	46	17.7 (0.8)	56.1% (45.6–69.1%)	<0.0001
	BRAF/MEKi	101	15	21.8 (0.6)	65.9% (48.4–89.7%)	
Overall survival (OS)						
	Overall	248	13	23.1 (0.3)	86.7% (78.8–95.3%)	
Indication group	LND/Primary	130	9	22.6 (0.4)	85.1% (74.6–97%)	0.1128
	M1	23	2	22.5 (1)	68.2% (37.6–100%)	
	Scar/In transit/LND	95	2	23.7 (0.2)	92.9% (83.3–100%)	
Adjuvant group	Anti-PD1	147	8	22.9 (0.4)	85.3% (74.9–97.2%)	0.6318
	BRAF/MEKi	101	5	23.2 (0.4)	87.8% (76.2–100%)	
Distant-metastases-free survival (DMFS)						
	Overall	248	49	20.3 (0.5)	70.2% (61.3–80.4%)	
Indication group	LND/Primary	130	22	20.7 (0.6)	72.4% (60.5–86.7%)	0.0525
	M1	23	8	16.7 (2.1)	60.9% (41.8–88.7%)	
	Scar/In transit/LND	95	19	20.6 (0.7)	70.2% (56.4–87.3%)	
Adjuvant group	Anti-PD1	147	37	18.9 (0.7)	64.8% (54.4–77.2%)	0.003
	BRAF/MEKi	101	12	22.2 (0.5)	76.5% (61.2–95.5%)	

* restricted mean calculated for the 24-month time span; ** occurrence of the following events is listed: for RFS: relapses, for OS: deaths, for DMFS: distant metastases; anti-PD1: nivolumab or pembrolizumab; BRAF/MEKi: dabrafenib plus trametinib; LND: lymph node dissection.

## Data Availability

The data presented in this study are available in this article.

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
