# Peer review of "Modern Approach to Melanoma Adjuvant Treatment with Anti-PD1 Immune Check Point Inhibitors or *BRAF/MEK* Targeted Therapy: Multicenter Real-World Report"

_cancers, 2023, doi:10.3390/cancers15174384_

Round 1

Reviewer 1 Report

Dear Authors; I found this work an interesting multicenter real-world report on "Modern approach to melanoma adjuvant treatment with anti- PD1 immune check point inhibitors or BRAF/MEK targeted 3 therapy". It needs some extra work for further processing it. Regards. P.S.

[1] Writing:

1-1 Expand introduction. It is too short. All three paragraphs need expansion. Also it does need an extra paragraph to showcase the readers of the paper outline.

1-2 Add list of used abbreviations in the work at the end of text right before Reference Section. Something like this: Abbreviations: OS: Overall Survival; ....

1-3 Discussion: Hard to read and follow it. Break it down to some subsections to improve smooth readership: 4.Discussion. 4.1. Summary & Contributions; 4.2. Strengths & Limitations; 4.3. Future Work

[2] Statistical

2-1 R Software used in line 140 needs citation. Also, mention which packages you used in your analysis and cite them .

Citation:

R Core Team (2021). R: A language and environment for statistical computing. R Foundation for Statistical Computing, Vienna, Austria.URL https://www.R-project.org/. 

2-2 You used multicenter data. I don't see where in your analysis you considered the center variation issue in your statistical analysis. This serious issue has been solved via meta-analysis methods in chapter 8 of the following reference. Hence, in one added subsection, it needs to be added to the manuscript revision for all overall outcomes of interest.  

Citation:

Chen, D.-G.(., Peace, K.E., & Zhang, P. (2017). Clinical Trial Data Analysis Using R and SAS (2nd ed.). Chapman and Hall/CRC. https://doi.org/10.1201/9781315155104

Author Response

Dear Reviewer,

Thank you for your insight and comments. Corrections were applied accordingly.

Point [1] Writing:

Point 1-1: Expand introduction. It is too short. All three paragraphs need expansion. Also it does need an extra paragraph to showcase the readers of the paper outline.

Response 1-1: Thank you for this comment, the introduction was expanded according to your guidance and paragraph on the paper outline added.

Point 1-2: Add list of used abbreviations in the work at the end of text right before Reference Section. Something like this: Abbreviations: OS: Overall Survival; ....

Response 1-2:  Added according to your recommendation.

Point 1-3: Discussion: Hard to read and follow it. Break it down to some subsections to improve smooth readership: 4. Discussion. 4.1. Summary & Contributions; 4.2. Strengths & Limitations; 4.3. Future Work

Response 1-3:  Corrected; discussion was subdivided.

Point [2] Statistical

Point 2-1: R Software used in line 140 needs citation. Also, mention which packages you used in your analysis and cite them.

Response 2-1:  The citation: “R Core Team (2021). R: A language and environment for statistical computing. R Foundation for Statistical Computing, Vienna, Austria.URL https://www.R-project.org/.” and the key packages were added.

Point 2-2: You used multicenter data. I don't see where in your analysis you considered the center variation issue in your statistical analysis. This serious issue has been solved via meta-analysis methods in chapter 8 of the following reference. Hence, in one added subsection, it needs to be added to the manuscript revision for all overall outcomes of interest.  

Citation: Chen, D.-G.(., Peace, K.E., & Zhang, P. (2017). Clinical Trial Data Analysis Using R and SAS (2nd ed.). Chapman and Hall/CRC. https://doi.org/10.1201/9781315155104

Response 2-2:  The idea behind our study was to exhaustively include the initial cohort of melanoma patients treated with adjuvant in Poland since this therapeutic option became routinely (i.e., not in clinical trials) available in 2019 until the end of study inclusion period in January 2021. In consequence, while we collected data from different centers, we tended to consider our results as real-world treatment outcomes in a single national cohort. In addition, we did not note statistically significant differences between the survivals of patients coming from different centers (Appendix B, Fig. B1). For these reasons we resolved to pool data for analysis. However, we agree that the variability across centers can potentially affect the findings, especially in prognostic models and we performed an additional analysis using the shared frailty Cox model to further investigate factors related to RFS and the impact of adding the random effect of the site. Indeed, the frailty term was insignificant in the model with small variance (Appendix A, Table A4).

Reviewer 2 Report

The authors presented important real world data on adjuvant chemotherapy for malignant melanoma. However, I believe that there is room for improvement in the method of analysis and I would like to request the following modifications.

1. Resected local recurrence (scar/in transit/nodal) is included as the 38.3% in the population; LND/Primary, M1, and Scar/In transit/LND are different populations, and lumping them all together is meaningless prognostic information. Therefore, Figure 1 should be omitted. In addition, prognostic analysis by the presence or absence of TRAEs also requires data for each population.

2. I would like to see the difference in treatment dose intensity (number of doses, dose reduction) between the TRAE group and the non-TRAE group, and I would like you to compare the difference in treatment content between the two groups by using indices such as dose intensity.

3. Figure B4 shows the survival curves by BRAF mutation status and treatment category, and it is very important information whether BRAF mutation patients should be treated with anti-PD-1 antibody or BRAF/MEK inhibitor as adjuvant therapy. The survival curve data should be included in the main figure, not in the Appendix, and a discussion of which drug should be used in BRAF mutations should be included in the main text.

Author Response

Dear Reviewer,

Thank you for your insight and comments. Corrections were applied accordingly.

Point 1: Resected local recurrence (scar/in transit/nodal) is included as the 38.3% in the population; LND/Primary, M1, and Scar/In transit/LND are different populations, and lumping them all together is meaningless prognostic information. Therefore, Figure 1 should be omitted.

Response 1:  We would prefer to retain the above-mentioned figure. The idea behind our study was to exhaustively include the initial cohort of melanoma patients treated with adjuvant in Poland since this therapeutic option became routinely (i.e., not in clinical trials) available in 2019 until the end of study inclusion period in January 2021. The composition of our cohort in terms of LND/Primary, M1, and Scar/In transit/LND populations is therefore representative of melanoma patients given adjuvant treatment in Poland and the overall result has epidemiological interpretation in our country, although admittedly it may not be directly applicable to other areas, where the relative proportions of LND/Primary, M1, and Scar/In transit/LND populations may be different. These three groups accurately reflect situations encountered in our daily practice and these cohorts were included in clinical trials and they are registration indications for aduvant therapy.

We added Table B5 in the Appendix to illustrate the effect of TRAE in each of the subpopulations separately. The trends are similar; however, we note that we do not have sufficient sample size to explore the survivals in the stratified analysis, especially in M1 group. We also investigated an interaction term in Cox proportional hazard model with occurrence of TRAE (Y/N) and the subpopulations (LND/Primary, M1, and Scar/In transit/LND), which was not statistically significant neither for overall survival nor for relapse free survival. We therefore concluded that we do not have sufficient evidence to indicate differential effect of TRAE depending on the subpopulation.

Point 2: I would like to see the difference in treatment dose intensity (number of doses, dose reduction) between the TRAE group and the non-TRAE group, and I would like you to compare the difference in treatment content between the two groups by using indices such as dose intensity.

Response 2:  Very intriguing question indeed.  Analysis of TRAE was not the focus of the current analysis and we did not collect sufficient data to be able to fully investigate the topic. In particular precise data on dosing may be prone to errors due to the retrospective nature of the study. For example, in Poland we cannot rely on the system when assessing an adequate use of tablets; we can only track dispensed, not actually used tablets and there is no such data recorded. Concerning immunotherapy- there were data collected only on the number of cycles and their planned, not actual, frequency and dosing. Those frequencies could and sometimes have changed and it is impossible to track them at this point. Unfortunately, data on actual dosage density was therefore not presented for the purpose of this study. While we agree that the topic merits further analysis, for completeness we summarized the available data in the Appendix C.

Point 3: Figure B4 shows the survival curves by BRAF mutation status and treatment category, and it is very important information whether BRAF mutation patients should be treated with anti-PD-1 antibody or BRAF/MEK inhibitor as adjuvant therapy. The survival curve data should be included in the main figure, not in the Appendix, and a discussion of which drug should be used in BRAF mutations should be included in the main text.

Response 3:  Thank you for the insight. Figure B4 was moved from Appendix to main text, numbers adjusted accordingly in figures and text; discussion included in the main text.

Reviewer 3 Report

I have carefully reviewed the manuscript titled 'Modern approach to melanoma adjuvant treatment with anti-2 PD1 immune checkpoint inhibitors or BRAF/MEK targeted therapy - multicenter real-world report'.

Overall, I find this paper to be both engaging and well-written, presenting valuable insights into the contemporary management of melanoma. The authors have skillfully compiled a multicenter real-world dataset, shedding light on the efficacy and outcomes of two pivotal treatment modalities: anti-2 PD1 immune checkpoint inhibitors and BRAF/MEK targeted therapy. The comprehensive analysis employed in this study contribute significantly to our understanding of adjuvant treatments for melanoma.

While I am largely impressed with the manuscript, I would suggest a few minor revisions to further enhance its clarity and impact.

While this study primarily focused on evaluating the effectiveness of anti-PD1 agents and DT, I was initially intrigued by the potential inclusion of histopathological parameters that could provide additional insights into the observed outcomes. Histopathological factors such as tumor thickness, ulceration, and mitotic rate have long been recognized as significant prognostic indicators in melanoma. However, it is important to note that a separate dedicated study might be necessary to thoroughly explore these parameters. Nevertheless, incorporating such factors may bolster the clinical implications of this research and make it more relevant for practicing histopathologists and oncologists.

Furthermore, I was surprised to note the relatively limited representation of acral and mucosal melanomas within the analyzed cohort. Given the distinctive clinical and biological characteristics of these subtypes, their underrepresentation raises questions about their prevalence and treatment patterns in the real-world setting. While I recognize that the focus of this study was on adjuvant therapies, the paucity of data on acral and mucosal melanomas within the multicenter cohort highlights an area of potential future investigation.

Finally, I would like to bring attention to the overall structure of the Discussion, which may appear lengthy and disorganized in some parts. Specifically, I found that the paragraph from lines 466 to 487 seemed tangential and did not seamlessly integrate with the surrounding content. It could be beneficial to reposition this paragraph within a more appropriate context or incorporate its content into the broader discussion for a better flow of ideas. By addressing this concern, we can improve the cohesiveness and impact of the discussion section, thus enhancing clarity and readability throughout your manuscript.

I appreciate the authors' contribution to melanoma treatment research and look forward to seeing this paper published after these recommended adjustments.

Author Response

Dear Reviewer,

Thank you for your insight and comments. Corrections were applied accordingly.

Point 1:separate dedicated study might be necessary to thoroughly explore these parameters. Nevertheless, incorporating such factors may bolster the clinical implications of this research and make it more relevant for practicing histopathologists and oncologists.

Response 1: I do agree that inclusion of histopathological, genetic and molecular parameters for translational research is a very important part of current research and I hope we will be able to perform a dedicated study in the future.

Point 2: … surprised to note the relatively limited representation of acral and mucosal melanomas within the analyzed cohort. Given the distinctive clinical and biological characteristics of these subtypes, their underrepresentation raises questions about their prevalence and treatment patterns in the real-world setting. While I recognize that the focus of this study was on adjuvant therapies, the paucity of data on acral and mucosal melanomas within the multicenter cohort highlights an area of potential future investigation.

Response 2: This is a very interesting topic indeed. I had the privilege of participating in an international dedicated project on the subject, led by Australian researchers:

ESMO poster; 809P Outcomes of patients with resected stage III/IV acral or mucosal melanoma treated with adjuvant anti-PD-1 therapy.

Jacques, S.K. et al.; Annals of Oncology, Volume 33, S915 - S916

Point 3: …overall structure of the Discussion, which may appear lengthy and disorganized in some parts. Specifically, I found that the paragraph from lines 466 to 487 seemed tangential and did not seamlessly integrate with the surrounding content. It could be beneficial to reposition this paragraph within a more appropriate context or incorporate its content into the broader discussion for a better flow of ideas.

Response 3: Discussion was subdivided into sections, redesigned and paragraph from lines 466 to 487 was adjusted according to the suggestions.

Reviewer 4 Report

A study by Placzke et al retrospectively assesses whether adjuvant systemic treatment outcomes in Polish melanoma patients are comparable to the results grom international trials. Although the patient group is not large, the analysis od interesting. Clinical data are presented very clearly. The conclusions are supported by data presented. The manuscript is very well written, the reasons of the study are well justified. The references cited are properly selected.

Minor comment:

1. Gene names should be written in italics (BRAF). In addition, please clarify "BRAF status" im Tables.

Author Response

Dear Reviewer,

Thank you for your insight and a comment.

Point 1: Gene names should be written in italics (BRAF). In addition, please clarify "BRAF status" in Tables.

Response 1: Corrections were made accordingly.

Round 2

Reviewer 1 Report

Dear Authors; my comments were addresses satisfactorily. Regards.  

Reviewer 2 Report

I checked the the authors' corrections and approve them.